# Unraveling the Influence of the Electrolyte on the Polarization Resistance of Nanostructured La_0.6_Sr_0.4_Co_0.2_Fe_0.8_O_3-δ_ Cathodes

**DOI:** 10.3390/nano12223936

**Published:** 2022-11-08

**Authors:** Javier Zamudio-García, Leire Caizán-Juanarena, José M. Porras-Vázquez, Enrique R. Losilla, David Marrero-López

**Affiliations:** 1Departamento de Química Inorgánica, Universidad de Málaga, 29071 Málaga, Spain; 2Departamento de Física Aplicada I, Universidad de Málaga, 29071 Málaga, Spain

**Keywords:** nanostructured materials, SOFC, cathode, electrolyte, spray-pyrolysis

## Abstract

Large variations in the polarization resistance of La_0.6_Sr_0.4_Co_0.2_Fe_0.8_O_3-δ_ (LSCF) cathodes are reported in the literature, which are usually related to different preparation methods, sintering temperatures, and resulting microstructures. However, the influence of the electrolyte on the electrochemical activity and the rate-limiting steps of LSCF remains unclear. In this work, LSCF nanostructured electrodes with identical microstructure are prepared by spray-pyrolysis deposition onto different electrolytes: Zr_0.84_Y_0.16_O_1.92_ (YSZ), Ce_0.9_Gd_0.1_O_1.95_ (CGO), La_0.9_Sr_0.1_Ga_0.8_Mg_0.2_O_2.85_ (LSGM), and Bi_1.5_Y_0.5_O_3-δ_ (BYO). The ionic conductivity of the electrolyte has a great influence on the electrochemical performance of LSCF due to the improved oxide ion transport at the electrode/electrolyte interface, as well as the extended ionic conduction paths for the electrochemical reactions on the electrode surface. In this way, the polarization resistance of LSCF decreases as the ionic conductivity of the electrolyte increases in the following order: YSZ > LSGM > CGO > BYO, with values ranging from 0.21 Ω cm^2^ for YSZ to 0.058 Ω cm^2^ for BYO at 700 °C. In addition, we demonstrate by distribution of relaxation times and equivalent circuit models that the same rate-limiting steps for the ORR occur regardless of the electrolyte. Furthermore, the influence of the current collector material on the electrochemical performance of LSCF electrodes is also analyzed.

## 1. Introduction

Solid oxide fuel cells (SOFCs) are one of the most promising technologies for efficient power generation and hydrogen production [1]. The main challenges regarding the commercialization of these devices are the reduction of the operating temperature and the development and optimization of alternative cell components (electrolyte, electrodes and interconnectors) with the aim of reducing their production costs [2,3].

One of the most limiting factors to achieve highly efficient SOFC devices at intermediate temperatures (600–800 °C) is the high polarization resistances of the cathode due to the sluggish oxygen reduction reaction (ORR) compared to the hydrogen oxidation reaction (HOR). In order to overcome this drawback, new cathode materials have been investigated, such as Sm_0.5_Sr_0.5_CoO_3-δ_, La_0.6_Sr_0.4_CoO_3-δ_, (Pr,Gd)BaCo_2_O_5+δ_, or Ca_3_Co_4_O_9+δ_, achieving remarkable results at 600 °C compared to the traditional La_0.8_Sr_0.2_MnO_3-δ_ (LSM) cathode [4,5]. However, these cobalt-based electrodes suffer from several disadvantages, including high thermal expansion coefficients, chemical reactivity with the electrolyte and low phase stability after long-term operation [6,7,8]. In contrast, iron-cobalt based perovskites, such as La_0.6_Sr_0.4_Co_0.2_Fe_0.8_O_3-δ_ (LSCF), exhibit moderate thermal expansion while having a good electrocatalytic activity for ORR, which makes it one of the most promising cathodes for SOFCs [9,10]. 

It is evident that the performance of SOFC electrodes depends on their intrinsic bulk properties, which are related to the composition and crystal structure, since they affect the electrical conduction and electrocatalytic activity for ORR. In particular, a high ionic conductivity is required to extend the so-called triple phase boundary (TPB), where the electrochemical reactions occur, to the whole electrode surface. In this sense, the electrode microstructure also plays a critical role in affecting the electrode performance, since the gas transport and number of active sites depend on the porosity and surface area.

The pristine LSCF cathode has been widely prepared by different synthetic and deposition routes, such as screen-printing, tape-casting, spray-drying, infiltration, spray-pyrolysis and pulsed laser deposition [11,12,13]. The different preparation methods and sintering temperatures have resulted in electrodes with different microstructures and, consequently, the polarization resistance varies in a wide range from 0.08 to 122 Ω cm^2^ at 700 °C (Table 1). Furthermore, different rate-limiting steps for the ORR have been identified depending on the synthetic method and the electrolyte choice. In particular, improved transport properties at the electrode/electrolyte interface have demonstrated to enhance the electrochemical performance by extending the surface paths for electrochemical reactions [14]. In fact, previous studies on infiltrated cathodes demonstrated a great influence of the ionic conductivity and surface exchange coefficient (k_s_) of the electrolyte scaffold on the electrochemical performance of the electrodes [15,16,17]. These results suggest that single electrodes could also be influenced by the level of ionic conductivity of the electrolyte.

These findings reveal the necessity of elucidating the influence of the electrolyte on the electrochemical properties of LSCF to better understand the electrode processes involved in the ORR and thereby being able to improve its performance. For this reason, in this study, nanostructured LSCF cathodes were prepared by spray-pyrolysis deposition on different oxide ion conducting electrolytes at reduced temperature to ensure negligible reactivity at the electrode/electrolyte interface. The structural and microstructural characterizations were carried out by X-ray diffraction (XRD) and scanning electron microscopy (SEM), respectively. A complete electrochemical characterization was carried out by impedance spectroscopy (EIS) at different oxygen partial pressures. The impedance spectra were analyzed by equivalent circuit models and distribution of relaxation times (DRT) to evaluate the nature of the electrode response as a function of the electrolyte. Additionally, the influence of the current collector layer (Pt, Ag and Au) on the electrochemical performance of LSCF was also tested.

## 2. Materials and Methods

### 2.1. Materials Preparation

Pellets of the electrolyte materials, Zr_0.84_Y_0.16_O_1.92_ (YSZ, Tosoh), Ce_0.9_Gd_0.1_O_1.95_ (CGO, Rhodia, Frankfurt, Germany), and La_0.9_Sr_0.1_Ga_0.8_Mg_0.2_O_2.85_ (LSGM, Kceracell, Chubu-myeon, Republic of Korea), were prepared from commercial powders. The powders were compacted into disks of 10 mm and 1 mm of diameter and thickness, respectively, and then sintered at 1400 °C for 4 h in air. The Bi_1.5_Y_0.5_O_3-δ_ (BYO) pellets were prepared from freeze-dried precursor powders and sintered at 1000 °C for 15 min in air as described elsewhere [16]. The relative density of all pellets was above 97%.

The La_0.6_Sr_0.4_Co_0.2_Fe_0.8_O_3-δ_ (LSCF) cathode was deposited by spray-pyrolysis onto YSZ, CGO, LSGM and BYO electrolytes from an aqueous solution (0.02 mol L^−1^) by dissolving stoichiometric amounts of the corresponding metal nitrate salts: La(NO_3_)_3_·6H_2_O, Sr(NO_3_)_2_, Co(NO_3_)_3_·6H_2_O, and Fe(NO_3_)_3_·9H_2_O (Merck, purity above 99%) in Milli-Q water under continuous stirring. The electrolyte substrates were heated on an aluminum block at 250 °C and then the precursor solution was sprayed with a flow rate of 20 mL h^−1^ for 1 h on each pellet face to obtain symmetrical cells. More details about the spray-pyrolysis procedure can be found elsewhere [11]. After the deposition, the layers were calcined in a furnace in air at 700 °C for 1 h with a heating/cooling rate of 2 °C min^−1^.

### 2.2. Structural, Microstructural and Electrical Characterization 

The composition and structure of the materials were studied by X-ray powder diffraction (XRD) with an Empyrean PANalytical diffractometer (CuK_α1,2_ radiation). The XRD patterns were analyzed with the Highscore Plus and GSAS suite software (3.0.5, PANalytical, Almelo, The Netherlands) [36,37]. The morphology of the LSCF films was observed by scanning electron microscopy in a FEI-SEM (Helios Nanolab 650, Hillsboro, OR, USA).

The electrochemical properties of the symmetrical cells were investigated by impedance spectroscopy with a Solartron 1260 FRA (Solartron Analytical, Hampshire, UK) in the frequency range of 0.01–10^6^ Hz and an AC amplitude of 50 mV. The impedance spectra were collected as a function of the temperature on cooling (700–450 °C) and the oxygen partial pressure (10^−3^–1 atm) by using an electrochemical cell equipped with an oxygen sensor and pump [38]. The data were analyzed by distribution of relaxation times (DRT) and equivalent circuit models with the help of DRTtools (1.0, Matlab 7.2) and ZView software (2.9c, Scribner Associates, Southern Pines, NC, USA), respectively [39,40]. The influence of the current collector layer on the performance of LSCF electrodes was also investigated using Pt-paste (Metalor Marin, Switzerland), Ag-paste (Sigma-Aldrich St. Louis, MO, USA), and Au-paste (Metalor), which were similarly painted on both faces of the pellets and then calcined at 700 °C for 30 min.

## 3. Results

### 3.1. Phase Formation

The XRD patterns of the LSCF cathode, deposited by spray-pyrolysis at 250 °C onto different electrolytes and calcined at 700 °C for 1 h, are displayed in Figure 1. The XRD patterns were adequately refined by the Rietveld method in the $Pm\bar{3}m$ space groups for LSCF, Pbnm for LSGM and $Fm\bar{3}m$ for YSZ, CGO and BYO, obtaining a good fitting of the experimental data with low disagreement factors (R_wp_ = 5–6%) (Table 2). Moreover, additional diffraction peaks associated with secondary phases are not observed due to the rapid fabrication and low temperature used in the present work. This avoids the formation of undesired reaction products, such as SrZrO_3_ and La_2_Zr_2_O_7_, at the LSCF/YSZ interface [28,41].

It has to also be mentioned that LSCF and LSGM phases are not easily discernible in the pattern because their diffraction peaks appear overlapped, since the crystal structure and cell parameters are similar (Figure 1c). It is also worth noting that bulk LSCF usually crystallizes with a rhombohedral symmetry (s.g. $R\bar{3}c$). However, a cubic polymorph is stabilized at room temperature due to the different oxygen non-stoichiometry and surface free energy of a nanostructured LSCF cathode. A similar behavior was previously observed for related compositions, such as La_0.6_Sr_0.4_Co_0.8_Fe_0.2_O_3-δ_ and La_0.6_Sr_0.4_CoO_3-δ_ [42,43].

Regarding the cell parameters of LSCF, these remain practically unaltered, regardless of the electrolyte used, ranging between 3.8820(4) and 3.8883(4) Å (Table 2). This finding further confirms that compounds with the same crystal structure and cation composition are obtained. The average crystallite size of LSCF, calculated by the Scherrer’s equation, is also independent of the electrolyte, varying between 21 and 24 nm (Table 2).

### 3.2. Microstructure

SEM images of the LSCF layers deposited onto different electrolytes with a thickness of approximately 1 mm are shown in Figure 2. The electrode layers have a similar thickness of 8 ± 1 µm and exhibit good adhesion to the substrates without delamination or cracks despite the low sintering temperature used (700 °C). A porous microstructure is achieved due to incomplete decomposition of the precursors during the deposition process at 250 °C, which are removed during the post-thermal treatment at 700 °C, creating additional porosity [42]. It should also be noted that no substantial differences are observed between the electrode morphologies regardless of the electrolyte used (Figure 2a–d). In all cases, the nanostructured electrodes show a laminated morphology, which is significantly different to that observed for traditional screen-printed LSCF electrodes sintered between 1000 and 1200 °C [22,30].

Lowering the deposition temperature has additional benefits for the electrode performance because the cation interdiffusion between the cell layers is minimized, preventing the chemical reactivity at the electrode/electrolyte interface [44]. In a previous work, LSCF cathode was directly deposited by spray-pyrolysis on YSZ electrolyte and negligible degradation was observed at low temperature after long-term operation [28].

Moreover, the thermal strain gradients caused by the mismatch between the thermal expansion behavior of the LSCF electrode (14.8·10^−6^ K^−1^) and the electrolytes (10.3, 12.1, 11.5 and ~13.6·10^−6^ K^−1^ for YSZ, CGO, LSGM and BYO, respectively) may cause delamination or cracks, negatively affecting the performance of conventional electrodes prepared at high sintering temperatures [45,46,47]. However, this drawback is reduced in nanostructured electrodes with a large number of grain boundary interfaces [11].

### 3.3. Electrochemical Characterization

The electrical characterization was carried out by impedance spectroscopy in symmetrical cells at open circuit voltage using Pt as current collector.

The impedance spectra show similar features for all the electrolytes, where the asymmetric arc suggests the presence of different contributions to the overall polarization resistance (Figure 3). The impedance spectra were firstly analyzed by distribution of relaxation times (DRT) to distinguish the different processes involved in the ORR (Figure 3). Two main contributions are observed in the DRT spectra regardless the electrolyte composition. The high frequency contribution (HF), centered at ~10^3^ Hz, is usually attributed to charge transfer processes at the electrode/electrolyte interface [48,49], while the low frequency process (LF), located between 10 and 100 Hz, is assigned to charge transfer or oxygen dissociation processes occurring on the electrode surface [50,51]. The latter process is the main resistive contribution to the overall polarization resistance. A minor contribution is also detected at a very low frequency of ~1 Hz, labelled as D, which is related to gas-phase diffusion limitations into the electrode [48,49]. Similar findings have previously been observed by DRT analysis for commercial, nanostructured, and nanofiber LSCF electrodes [52].

Based on the DRT analysis, the impedance spectra were fitted using the equivalent circuit of the inset of Figure 3g, where *L* is an inductor attributed to the electrochemical setup and *R_Ω_* represents the electrolyte resistance. Each specific electrode process was fitted by considering an *RQ* element, where *R* is a resistance in parallel with a constant phase element *Q*. Since the diffusion process D is rather low, it was not considered during the fitting.

For a better understanding of the influence of the electrolyte on the ORR properties, each electrode contribution was studied separately as a function of temperature (Figure 4). It is found that the resistance of the high (R_HF_) and low frequency (R_LF_) contributions are affected by the composition of the electrolyte. In particular, R_HF_ decreases as the ionic conductivity of the electrolyte increases, YSZ > CGO ≥ LSGM > BYO, attributed to a faster oxide ion transfer at the electrode/electrolyte interface. This is further confirmed by the lower activation energy of BYO (1.08 eV) compared to CGO (1.14 eV), LSGM (1.25 eV) and YSZ (1.33 eV) (inset Figure 4a). Similarly, R_LF_ is strongly affected by the ionic conductivity of the electrolyte, indicating that the electrolyte not only improves the oxide-ion transfer at the interface, but also extends the surface paths for the electrochemical reaction [53]. Similar findings were observed for different active layers incorporated between the cathode and the electrolyte [14,53,54]. As expected, the corresponding activation energy values for R_LF_ are higher than those for R_HF_, ranging from 1.35 eV for BYO to 1.61 eV for YSZ (inset Figure 4b). 

In contrast, the capacitances of the high (C_HF_) and low (C_LF_) frequency contributions are nearly independent of the measuring temperature and the electrolyte (Figure 4c,d). It is well reported that the electrode capacitance increases for those electrodes with high TPB length [55]. For this reason, the LSCF nanostructured electrodes exhibit high capacitance values of ~6·10^−3^ and 2·10^−2^ F cm^−2^ for the HF and LF contributions, respectively. 

Figure 5 shows the overall polarization resistance (R_p_ = R_HF_ + R_LF_) as function of the temperature for LSCF deposited onto the different electrolytes. In the whole temperature range, R_p_ decreases as the ionic conductivity of the electrolyte increases in the following order: YSZ > CGO > LSGM > BYO. The LSCF deposited onto BYO shows an R_p_ value of only 0.058 Ω cm^2^ at 700 °C, which is one of the lowest values reported in the literature for pure La_0.6_Sr_0.4_Co_0.2_Fe_0.8_O_3-δ_ (Table 1). Similarly, the activation energy of R_p_ decreases from 1.48 eV for YSZ to 1.39 eV for BYO, which confirms that an increase of the ionic conductivity of the electrolyte induces faster ORR kinetics. 

In order to get further insight into the rate-limiting steps involved in the ORR, the impedance spectra were collected as a function of the oxygen partial pressure (pO_2_). The relationship between the electrode polarization resistance of each process and the oxygen partial pressure can be expressed as R = R_o_(pO_2_)^−*m*^, where the parameter *m* provides information about the nature of the electrode processes. Despite the fact that the complete ORR can be summarized as follows, $O_{2}+2V_{ö}+4e^{\prime }↔\;2O_{o}^{x}$, this reaction comprises multiple sub-steps with different *m* values that are displayed in Figure 6a: (1) the oxygen adsorption on the electrode surface (*m* = 1, $O_{2,gas}\to O_{2,\;ads}$); (2) the oxygen gas dissociation on the electrode surface (*m* = 1/2, *O*_2,*ads*_ → 2*O_ads_*); (3) one electron oxygen reduction (*m* = 3/8, $O_{ad}+e^{\prime }\to O_{ad}^{\prime }$) and (4) (*m* = 1/8, $O_{ad}^{\prime }+e^{\prime }\to O_{ad}^{\prime \prime }$) or (5) complete reduction of the oxygen atoms (*m* = 1/4, $O_{ad}+2e^{\prime }+V_{ö}\to O_{o}^{x}$); and (6) the oxygen ion incorporation at the electrode/electrolyte interface (*m* = 0, $O_{o,\mathrm{electrode}}^{x}\to O_{o,\mathrm{eletrolyte}}^{x}$) [56]. It has to be noted that all these steps are not detectable for a specific electrode. Only the rate-limiting steps are discernible in the impedance spectra. In this context, different rate-limiting steps have been proposed in the literature for pure LSCF, obtained from different preparation routes and deposited onto different electrolytes (see Table 1), which difficulties the possibility to stablish a clear relationship between the electrolyte used and the rate-limiting steps involved in the ORR. It its worth mentioning that this is a critical factor to design an adequate electrode microstructure and to improve the cell performance and durability.

Figure 6b,c show representative impedance spectra at 700 °C and different oxygen partial pressures for LSCF deposited onto YSZ and CGO electrolytes, respectively. Since the closeness of the characteristic frequencies of each sub-reaction hinders the identification of the different electrode processes, DRT analysis of the impedance data was performed (Figure 6d,e). The DRT analysis confirms the presence of two main electrode contributions for all samples in the whole pO_2_ range studied. As expected, the higher the pO_2_ the lower overall electrode polarization resistance. R_HF_ and R_LF_ contributions were determined by the integral area under each peak and represented in Figure 7a,b, respectively. The HF process is nearly independent of the oxygen partial pressure for all samples (*m* ~ 0), confirming that this process is related to the oxide ion transport across the electrode/electrolyte interface. On the other hand, the LF process shows a *m* ~ 0.5 for CGO, LSGM, and BYO electrolytes, which is attributed to oxygen dissociation at the electrode surface [27]. However, a different rate-limiting-step is observed for LSCF deposited on YSZ. In this case, the *m* ~ 3/8 indicates that this process is possibly assigned to oxygen dissociative adsorption followed by charge transfer [55,57]. The poorer oxide ion conductivity of YSZ at intermediate temperatures, in comparison to CGO, LSGM, and BYO, could explain the different behavior. In fact, the ionic conduction in YSZ cell is limited at the electrode/electrolyte interface, inducing a slower charge transfer at the electrode surface. In the literature, different ORR processes were found, due to the different synthetic methods and sintering temperatures employed, which highly affect the electrode microstructure and thus the electrochemical properties (Table 1). For instance, Wang et al. [26] found two rate-limiting steps for screen-printed LSCF on Sm-doped CeO_2_ electrolyte with *m* = 0.24 and 0.44. Marinha et al. [21] observed three processes for spray-pyrolysis LSCF electrodes on CGO with *m* = 0, 0.28 and 0.71 whiles Kim et al. [23] observed only two processes with *m* = 0 and 0.28 (Table 1). Moreover, in contrast to previous works [45,47], gas diffusion limitations (*m* = 1) are not observed for the spray-pyrolysis electrodes (Table 1). Hence, the nanostructured LSCF electrodes, deposited by spray-pyrolysis at low temperature, have an adequate porosity, avoiding problems related to gas diffusion.

### 3.4. Influence on the Current Collector Layer

Finally, we have studied the effect of the current collector material on the electrochemical properties of the LSCF nanostructured electrode deposited onto a CGO electrolyte. It is well known that an adequate current collector is needed to ensure a uniform electric field and proper efficiency of the cell. Some current collectors, such as Pt, can be electrochemically active for ORR, or even react with the electrode [58]. For these reasons, Ag and Au pastes are also investigated as current collectors in a symmetrical cell configuration. Figure 8a compares the impedance spectra of LSCF/CGO/LSCF cells with different current collectors, where almost identical R_p_ values are observed for the Pt- and Ag-containing cells (~0.29 Ω cm^2^ at 650 °C). However, the Au-containing cell shows slightly higher values (0.34 Ω cm^2^). 

A deeper analysis of the different electrode contributions to the overall polarization resistance reveals that the HF processes, attributed to the oxide ion transfer at the electrode/electrolyte interface, are similar for all samples (Figure 8b). Thus, the main differences between the cells are related to changes in the LF response. In particular, the cell with Au current collector shows the highest R_LF_ values. SEM images of the current collector surface reveal a good porosity for both Pt and Ag cells. In contrast, a dense layer with low superficial porosity is observed for the Au current collector, which partially blocks the active sites on the electrode surface and, as consequence, the R_LF_ contribution increases (Figure 8c–e). A similar behavior was observed for La_0.8_Sr_0.2_FeO_3-δ_ electrode with Au current collector when compared to Pt and Ag [59]. 

## 4. Conclusions

Nanostructured La_0.6_Sr_0.4_Co_0.2_Fe_0.8_O_3-δ_ (LSCF) electrodes were prepared by spray-pyrolysis deposition onto different electrolytes: Zr_0.84_Y_0.16_O_1.92_ (YSZ), Ce_0.9_Gd_0.1_O_1.95_ (CGO), La_0.9_Sr_0.1_Ga_0.8_Mg_0.2_O_2.85_ (LSGM), and Bi_1.5_Y_0.5_O_3-δ_ (BYO). The electrochemical measurements revealed a direct relationship between the ionic conductivity of the electrolyte and the polarization resistance of the cells, which was improved by a faster oxide ion transport at the electrode/electrolyte interface and an extension of the surface paths for the electrochemical reactions. The polarization resistance of LSCF deposited onto BYO was 3.5 times lower than that observed for YSZ, i.e., 0.058 and 0.21 Ω cm^2^ at 700 °C, respectively, which was attributed to the poorer ionic conductivity of YSZ at intermediate temperatures. The analysis of the impedance spectra by distribution of relaxation times and equivalent circuit models revealed that similar rate-limiting steps are involved in the ORR for CGO, LSGM, and BYO electrolytes, which were assigned to oxygen gas dissociation on the electrode surface and oxide ion incorporation at the electrode/electrolyte interface. An additional study regarding the influence of the current collector layer revealed that both Pt and Ag are suitable materials to obtain representative polarization resistance values, while Au partially blocks the active sites for oxygen reduction on the electrode surface. All these results demonstrate that the ionic conductivity at the electrode/electrolyte interface is a bottleneck for the electrode performance of LSCF, results that can be extended to other electrodes for SOFCs.

## Figures and Tables

**Figure 1 nanomaterials-12-03936-f001:**
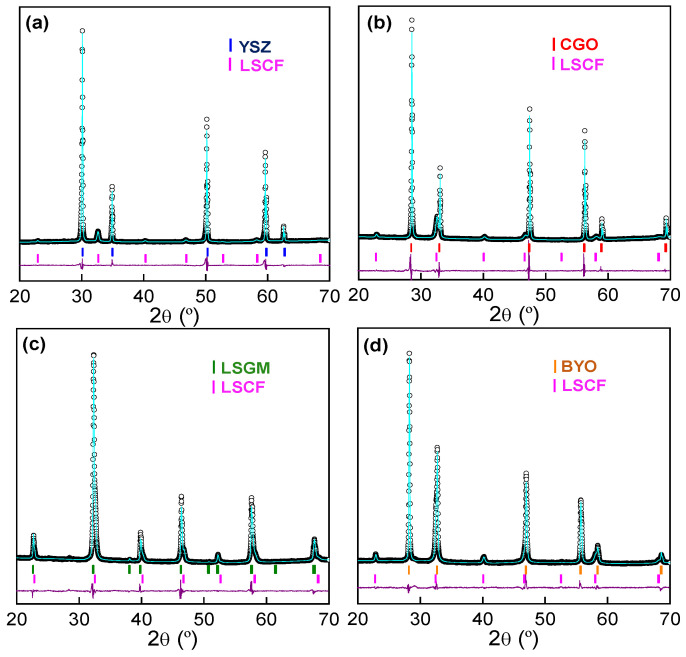
Rietveld refinement of LSCF electrode deposited onto different electrolytes at 700 °C, (**a**) Zr_0.84_Y_0.16_O_1.92_ (YSZ), (**b**) Ce_0.9_Gd_0.1_O_1.95_ (CGO), (**c**) La_0.9_Sr_0.1_Ga_0.8_Mg_0.2_O_2.85_ (LSGM) and (**d**) Bi_1.5_Y_0.5_O_3_ (BYO).

**Figure 2 nanomaterials-12-03936-f002:**
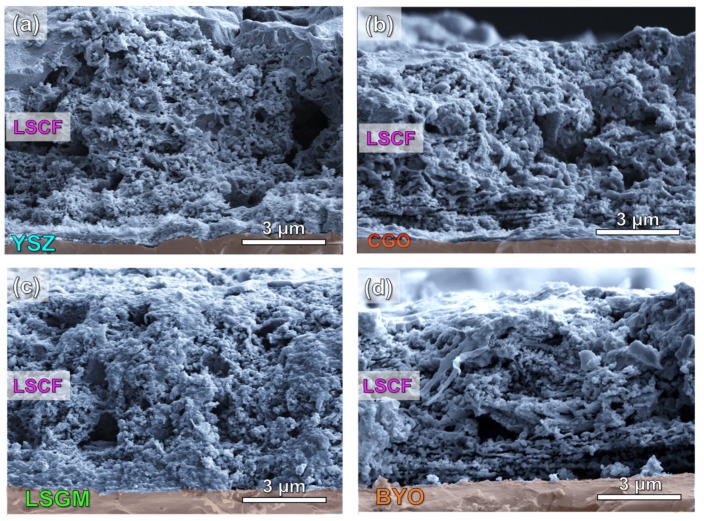
SEM images of the LSCF deposited on different electrolytes at 700 °C, (**a**) YSZ, (**b**) CGO, (**c**) LSGM and (**d**) BYO.

**Figure 3 nanomaterials-12-03936-f003:**
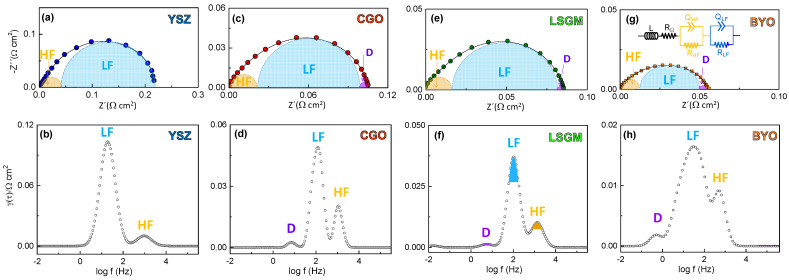
Impedance spectra (top) and DRT analysis (bottom) at 700 °C of the LSCF electrode deposited on (**a**,**b**) YSZ, (**c**,**d**) CGO, (**e**,**f**) LSGM and (**g**,**h**) BYO. The inset of (**g**) shows the equivalent circuit used to fitting the data.

**Figure 4 nanomaterials-12-03936-f004:**
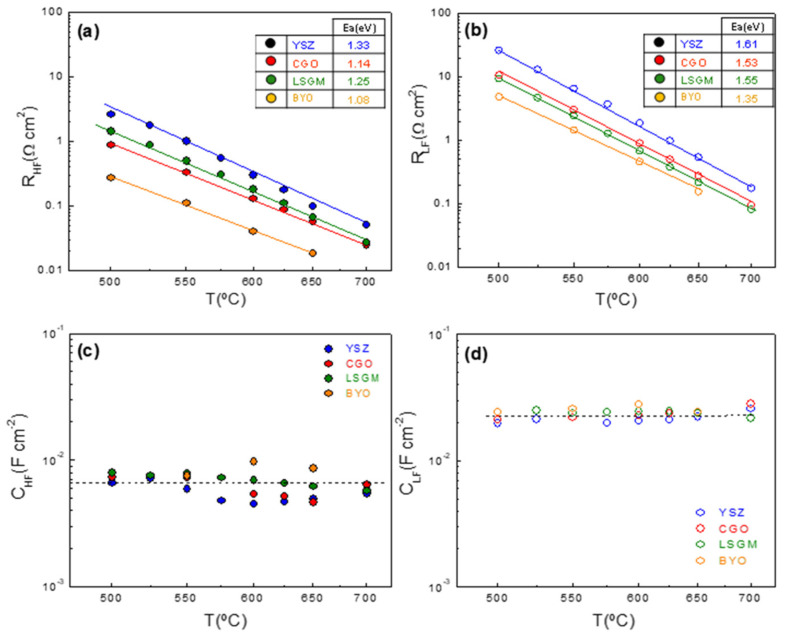
Temperature dependence of (**a**) R_HF_ and (**b**) R_LF_ resistance and (**c**) C_HF_ and (**d**) C_LF_ capacitance contributions of LSCF electrodes deposited on different electrolytes.

**Figure 5 nanomaterials-12-03936-f005:**
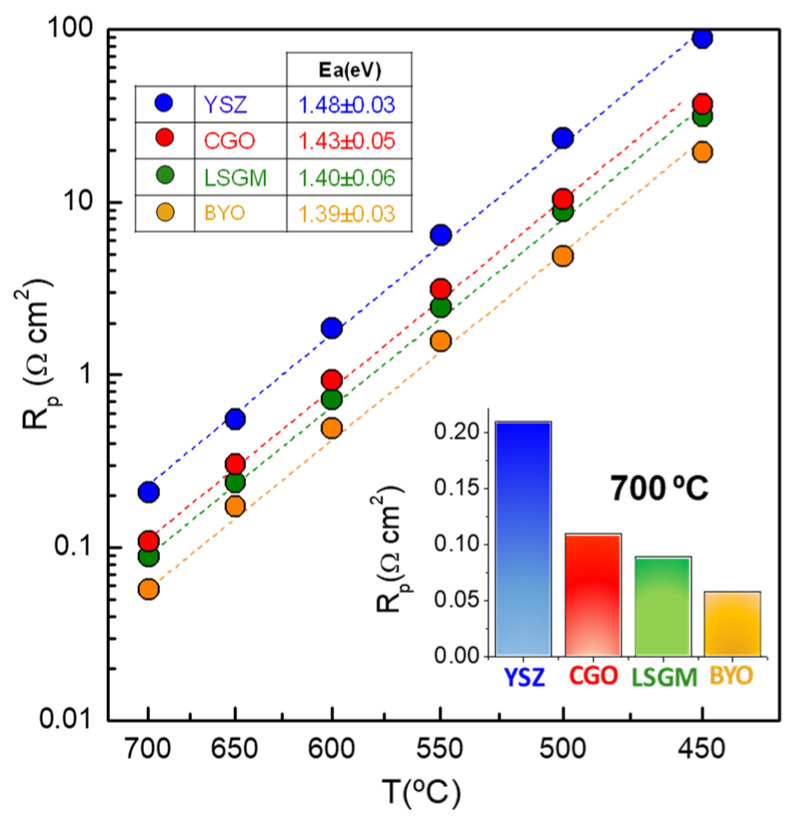
Overall polarization resistance of LSCF nanostructured electrodes deposited onto different electrolytes as a function of temperature. The inset figure compares the polarization resistance at 700 °C.

**Figure 6 nanomaterials-12-03936-f006:**
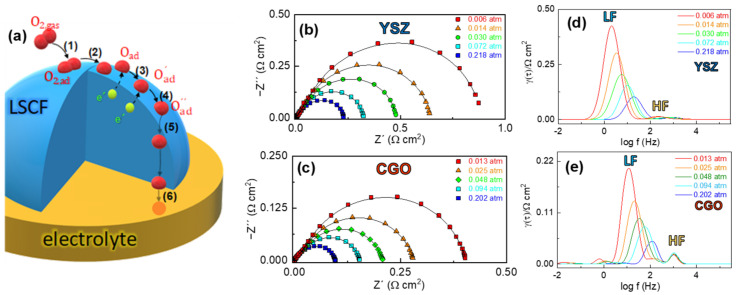
(**a**) Schematic diagram of the different sub-reactions of the oxygen reduction reaction. Impedance spectra at 700 °C of (**b**) LSCF/YSZ/LSCF and (**c**) LSCF/CGO/LSCF cells as a function of the oxygen partial pressure and the corresponding (**d**,**e**) DRT plots.

**Figure 7 nanomaterials-12-03936-f007:**
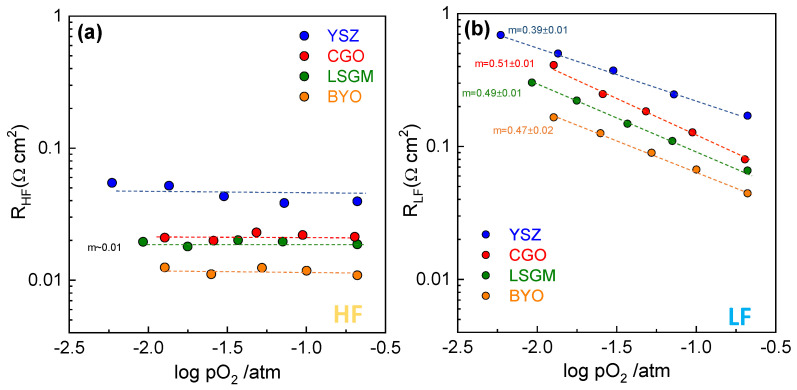
Polarization resistances of the (**a**) HF and (**b**) LF electrode contributions of LSCF with different electrolytes as a function of the oxygen partial pressure at 700 °C.

**Figure 8 nanomaterials-12-03936-f008:**
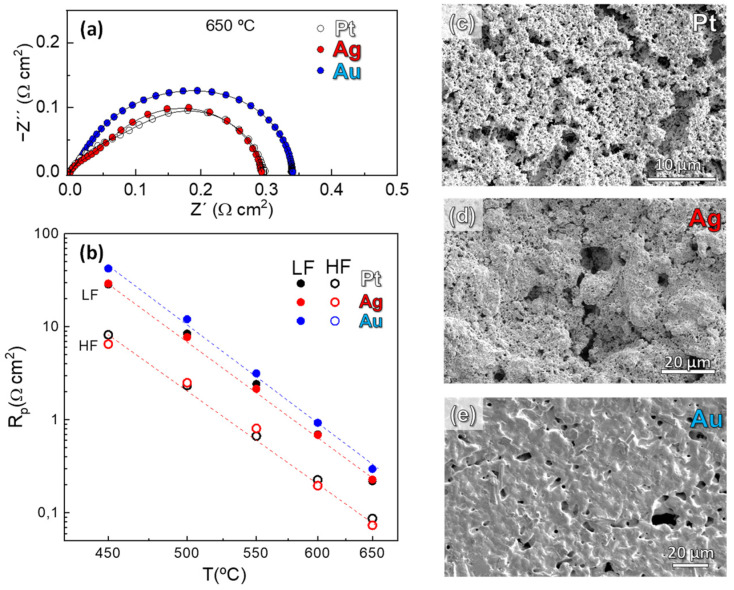
(**a**) Impedance spectra at 650 °C of LSCF electrodes deposited onto CGO electrolyte with Pt, Ag and Au as current collectors and (**b**) electrode resistance contributions as a function of the temperature. SEM images of the surface of (**c**) Pt, (**d**) Ag and (**e**) Au current collectors after the electrochemical tests.

**Table 1 nanomaterials-12-03936-t001:** Brief summary of the polarization resistances (R_p_) and reaction order (*m*) of the rate-limiting steps of the ORR of La_0.6_Sr_0.4_Fe_0.8_Co_0.2_O_3-δ_ (LSCF) at 700 °C reported in the literature. The temperature is included when data is not available at 700 °C.

Fabrication Technique	Electrolyte	R_p_ (Ω cm^2^)	*m*	Ref.
Spray-pyrolysis	Zr_0.84_Y_0.16_O_1.92_	0.21	~0 0.39	This work
Spray-pyrolysis	Ce_0.9_Gd_0.1_O_1.95_	0.11	~0 0.51	This work
Spray-pyrolysis	La_0.9_Sr_0.1_Ga_0.8_Mg_0.2_O_3-δ_	0.089	~0 0.49	This work
Spray-pyrolysis	Bi_1.5_Y_0.5_O_3_	0.058	~0 0.47	This work
Spray-pyrolysis	Ce_0.8_Gd_0.2_O_1.9_	0.7^650 °C^	-	[18]
Spray-pyrolysis	Ce_0.8_Gd_0.2_O_1.9_	0.9	-	[19]
Freeze-Drying	Ce_0.8_Gd_0.2_O_1.9_	0.3	-	[20]
Spray-pyrolysis	Ce_0.8_Gd_0.2_O_1.9_	10^600 °C^	0 0.20 0.71	[21]
Screen-printing	Ce_0.9_Gd_0.1_O_1.95_	0.12	-	[22]
Solid-state reaction	Ce_0.9_Gd_0.1_O_1.95_	15^750 °C^	0 0.28	[23]
Magnetron sputtering	Ce_0.9_Gd_0.1_O_1.95_	122	-	[24]
Screen-printing	Ce_0.9_Gd_0.1_O_1.95_	0.21^750 ºC^	-	[25]
Screen-printing	Sm_0.2_Ce_0.8_O_1.9_	0.55^600 °C^	0.24 0.44	[26]
Co-precipitation	BaZr_0.8_Y_0.2_O_3-δ_	0.70^600 °C^	0.16 0.50	[27]
Spray-pyrolysis	Zr_0.84_Y_0.16_O_1.92_	0.3^650 °C^	-	[28]
Spray-pyrolysis	Zr_0.84_Y_0.16_O_1.92_	3^600 °C^	-	[29]
Spray-drying	Zr_0.84_Y_0.16_O_1.92_	2	-	[30]
Spin coating	Zr_0.84_Y_0.16_O_1.92_	0.1^750 °C^	-	[31]
Pulsed laser deposition	Zr_0.84_Y_0.16_O_1.92_	9^750 °C^	-	[13]
Tape casting	Zr_0.84_Y_0.16_O_1.92_	0.4	-	[32]
Spray-drying	Zr_0.84_Y_0.16_O_1.92_	5	~0 0.25	[33]
Spray-drying	La_0.9_Sr_0.1_Ga_0.8_Mg_0.2_O_3-δ_	0.19	-	[34]
Screen-printing	La_0.8_Sr_0.2_Ga_0.8_Mg_0.2_O_3-δ_	0.08	0.10 0.82	[35]

**Table 2 nanomaterials-12-03936-t002:** Structural and microstructural parameters of LSCF cathode deposited onto different electrolytes by spray-pyrolysis.

Electrolyte	Lattice Parameter (Å)	Cell Volume (Å^3^)	R_wp_ (%)	d_LSCF_ (nm)
YSZ	3.8828 (4)	58.540 (2)	4.57	24
CGO	3.8876 (4)	58.540 (2)	4.86	20
LSGM	3.8820 (4)	58.540 (2)	4.01	-
BYO	3.8883 (4)	58.540 (2)	3.01	22

## Data Availability

The data presented in this study are available on request from the corresponding author.

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
