# Peer review of "Unraveling the Influence of the Electrolyte on the Polarization Resistance of Nanostructured La0.6Sr0.4Co0.2Fe0.8O3-δ Cathodes"

_nanomaterials, 2022, doi:10.3390/nano12223936_

Round 1

Reviewer 1 Report

Unraveling the influence of the electrolyte on the polarization resistance of nanostructured La0.6  Sr0.4Co0.2Fe0.8O3‐d cathodes

An investigation of the effect of different electrolytes on the polarization resistance of nanostructured LSCF 6428 was presented in this manuscript. As a result, this work looks interesting as it will help the scientific community to understand how electrolyte-electrode interfaces affect polarization resistance. It is clear and convincing how the results were obtained, and I have a few suggestions. After some clarification, this work could be accepted.

1.        In Figure 2, the electrolyte and electrode thickness are not indicated, which makes the comparison unfair.

2.        In the XRD position of LCSF, there is a difference in intensity in figure 1. Just curious about how to control LSCF6428 deposition on electrolyte distribution and concentration.

3.      To compare the composition of LSCF6428 on different electrolytes, add an EDX scan of the FESEM image, if possible.

4.     The XRD of the bare LSCF6428 particle without electrolyte after calcining at 700°C is still intriguing to me. I am concerned about phase purity. Would you be able to provide an XRD for it or have you checked it on XRD. It's just out of curiosity and for my own knowledge, nothing serious.

Author Response

An investigation of the effect of different electrolytes on the polarization resistance of nanostructured LSCF 6428 was presented in this manuscript. As a result, this work looks interesting as it will help the scientific community to understand how electrolyte-electrode interfaces affect polarization resistance. It is clear and convincing how the results were obtained, and I have a few suggestions. After some clarification, this work could be accepted.

We thank the reviewer for the useful comments and revisions, which are considered for the improvement of the manuscript. The changes are highlighted in the manuscript.

  1. In Figure 2, the electrolyte and electrode thickness are not indicated, which makes the comparison unfair.

Response 1: The thicknesses of the electrolyte ~1 mm and the electrodes 8±1 µm are commented in the manuscript on page 4 (highlighted text).

  1. In the XRD position of LCSF, there is a difference in intensity in figure 1. Just curious about how to control LSCF6428 deposition on electrolyte distribution and concentration.

Response 2: The layers are simultaneously deposited onto electrolytes with different composition (YSZ, CGO, BYO) or crystal structure (LSGM). The diffraction peak intensity of the different electrolyte substrates depends on the composition, and therefore, the relative intensity of LSCF is different in the patterns of Fig. 1. The spray-pyrolysis process for a related composition LSCF6428 was described in details elsewhere [25]. A precursor solution is atomized and deposited on a heated substrate, which is continuously moving under the spray nozzle to ensure a homogenous layer distribution. The morphology (porosity and thickness) is tailored by varying the deposition temperature and time. A detailed analysis of the phase composition and crystal structure are reported in this previous work [25].

  1. To compare the composition of LSCF6428 on different electrolytes, add an EDX scan of the FESEM image, if possible.

Response 3: The composition of LSCF on the different electrolytes was confirmed by XRD Rietveld method (page 3). Table 2 compares the lattice and volume cell of LSCF on different electrolytes. The lattice cell is practically the same, ranging between 3.8820(4) and 3.8883(4) Å, indicating that the cation composition and crystal structure are similar. The cation composition of the different LSCF layers cannot be correctly determined by EDX-FESEM technique because the particle size of LSCF is about 20-25 nm and the interaction volume of the characteristic X-ray by SEM technique is very high and the particles cannot be differentiated. Notice that we have used the same cation precursor solution for the preparation of all layers but minor phase segregations cannot be detected by conventional SEM-EDX. 

  1. The XRD of the bare LSCF6428 particle without electrolyte after calcining at 700°C is still intriguing to me. I am concerned about phase purity. Would you be able to provide an XRD for it or have you checked it on XRD. It's just out of curiosity and for my own knowledge, nothing serious.

Response 4: In this study, LSCF6428 was directly deposited on the electrolyte by spray-pyrolysis deposition and the XRD peaks of LSCF are not are not easily distinguishable. In a previous work [25], we prepared a related composition LSCF6482 by spray-pyrolysis on amorphous quartz substrates and the patterns are compared with those of powder samples obtained by citrate precursor method. We have confirmed that the same crystal and composition were obtained. Please, find attached below the XRD patterns of our previous work. In this study the phase purity was confirmed by XRD Rietveld method.

Reviewer 2 Report

In this work titled „ Unraveling the influence of the electrolyte on the polarization resistance of nanostructured La0.6Sr0.4Co0.2Fe0.8O3 cathodes”, the polarizations of LSCF cathodes deposited on different electrolytes: Zr0.84Y0.16O1.92 (YSZ), Ce0.9Gd0.1O1.95 (CGO), La0.9Sr0.1Ga0.8Mg0.2O2.85 (LSGM) and Bi1.5Y0.5O3 (BYO) have been evaluated. The analysis of the impedance spectra by distribution of relaxation times and equivalent circuit models was conducted. In addition, the chemical compatibility of LSCF cathode towards YSZ, CGO, LSGM and BYO electrolytes was analyzed after the annealing at 700 ºC for 1 h. The microstructure has also been investigated by SEM. In the final, the influence of the current collector material on the electrochemical performance of LSCF electrodes was also analyzed.

In general, the work is of interest for the readers and can be published in the Nanomaterials after proper modifications. The following suggestions can be taken into consideration for the further improvement.

1.      The influence of thermal expansion coefficient (TEC) mismatch between LSFC and YSZ, CGO, LSGM and BYO electrolytes can be included in the work.

2.      The aspect of chemical compatibility of LSFC with YSZ, CGO, LSGM and BYO electrolytes should be explored. It is well known that LSCF reacts with Co-containing materials, such as LSCF. Therefore, it is highly recommended to conduct EDS elemental mapping of the interface between LSFC and electrolytes.

3.      The authors may carefully recheck the whole manuscript and correct linguistic errors.

Author Response

In this work titled „ Unraveling the influence of the electrolyte on the polarization resistance of nanostructured La0.6Sr0.4Co0.2Fe0.8O3 cathodes”, the polarizations of LSCF cathodes deposited on different electrolytes: Zr0.84Y0.16O1.92 (YSZ), Ce0.9Gd0.1O1.95 (CGO), La0.9Sr0.1Ga0.8Mg0.2O2.85 (LSGM) and Bi1.5Y0.5O3 (BYO) have been evaluated. The analysis of the impedance spectra by distribution of relaxation times and equivalent circuit models was conducted. In addition, the chemical compatibility of LSCF cathode towards YSZ, CGO, LSGM and BYO electrolytes was analyzed after the annealing at 700 ºC for 1 h. The microstructure has also been investigated by SEM. In the final, the influence of the current collector material on the electrochemical performance of LSCF electrodes was also analyzed.

In general, the work is of interest for the readers and can be published in the Nanomaterials after proper modifications. The following suggestions can be taken into consideration for the further improvement.

We thank to the reviewer for the useful suggestions, which are considered to improve the manuscript. The changes are highlighted in the manuscript.

  1. The influence of thermal expansion coefficient (TEC) mismatch between LSFC and YSZ, CGO, LSGM and BYO electrolytes can be included in the work.

Response 1: We have commented on page 4 the influence of the TEC values on the performance of SOFC electrodes. Adequate references are also included in the manuscript [30-32].  

“Moreover, the thermal strain gradients caused by the mismatch between the thermal expansion behavior of the LSCF electrode (14.8·10-6 K-1) and the electrolytes (10.3, 12.1, 11.5 and ~13.6·10-6 K-1 for YSZ, CGO, LSGM and BYO, respectively) may cause delamination or cracks, negatively affecting the electrode performance of conventional electrodes prepared at high sintering temperatures [30–32]. However, this drawback is reduced in nanostructured electrodes with a large number of grain boundary interfaces [11]”.

  1. The aspect of chemical compatibility of LSFC with YSZ, CGO, LSGM and BYO electrolytes should be explored. It is well known that LSCF reacts with Co-containing materials, such as LSCF. Therefore, it is highly recommended to conduct EDS elemental mapping of the interface between LSFC and electrolytes.

Response 2: It is well reported in the literature that LSCF is chemically incompatible with YSZ electrolyte due to the formation of SrZrO3 and La2Zr2O7 reaction products at the electrode/electrolyte interface during the fabrication process of the cell at high temperature. A protective CeO2 based layer is usually employed to prevent this drawback. In the present study, the LSCF layers were deposited on different electrolytes at only 450 ºC and the sintering and working temperature were limited at 700 ºC to prevent the formation of undesired reaction products. In a previous work [23], LSCF cathodes were deposited directly on the YSZ electrolyte and negligible degradation was observed at low temperature after long-term operation; however, in the present work the long-term stability of the cell was not investigated because this is not the aim of the present manuscript. These comments are included in the manuscript on page 3 and 4. On the other hand, LSCF is chemically compatible with CGO, LSGM and BYO at the working temperatures used in the present study [29]. Furthermore, EDS mapping by SEM is not an useful technique to study the chemical compatibility at the electrode/electrolyte interface of thin films because the volume interaction of characteristic X-rays is relatively high ~ 1 µm, compared to the particle size, and therefore, the preparation of lamellas and TEM studies are required, which is outside the scope of this paper.

  1. The authors may carefully recheck the whole manuscript and correct linguistic errors

Response 3: The whole manuscript has been carefully revised and minor grammatical errors are corrected.

Round 2

Reviewer 2 Report

Accept in present form